# Locally Differentially Private (Contextual) Bandits Learning

**Kai Zheng**[1]
zhengk92@gmail.com

**Tianle Cai**[2,3]
caitianle1998@pku.edu.cn

**Weiran Huang**[4]
weiran.huang@outlook.com

**Zhenguo Li**[4]
li.zhenguo@huawei.com

**Liwei Wang**[5,6,*]
wanglw@cis.pku.edu.cn

## Abstract

We study locally differentially private (LDP) bandits learning in this paper. First, we propose simple black-box reduction frameworks that can solve a large family of context-free bandits learning problems with LDP guarantee. Based on our frameworks, we can improve previous best results for private bandits learning with one-point feedback, such as private Bandits Convex Optimization, and obtain the first result for Bandits Convex Optimization (BCO) with multi-point feedback under LDP. LDP guarantee and black-box nature make our frameworks more attractive in real applications compared with previous specifically designed and relatively weaker differentially private (DP) context-free bandits algorithms. Further, we extend our $(\varepsilon, \delta)$-LDP algorithm to Generalized Linear Bandits, which enjoys a sub-linear regret $\tilde{\mathcal{O}}(T^{3/4}/\varepsilon)$ and is conjectured to be nearly optimal. Note that given the existing $\Omega(T)$ lower bound for DP contextual linear bandits [35], our result shows a fundamental difference between LDP and DP contextual bandits learning.

## 1 Introduction

As a general and powerful model, (contextual) bandits learning has attracted lots of attentions both in theoretical study and real applications [8, 28], from personalized recommendation to clinical trails. However, existing algorithms designed for these applications heavily rely on user's sensitive data, and an off-the-shelf use of such algorithms may leak user's privacy and bring concerns to future users for sharing their data with related institutions or corporations. For example, in classification or regression tasks, we update our model according to the feature and label of each user. In Multi-Armed Bandits (MAB), we estimate underlying rewards of all arms based on user's feedback. A solid notion of data privacy is Differential Privacy (DP) proposed by Dwork et al. [13] in 2006. Since then, differentially private bandits learning has been studied extensively.

Among context-free bandits learning, Bandits Convex Optimization (BCO) is one of the fundamental problems. Thakurta and Smith [37] designed the first $(\varepsilon, \delta)$-*differentially private* adversarial BCO

[*] Corresponding author.

| Type | Problem | | Our Regret Bound | Best Non-Private Regret |
|---|---|---|---|---|
| Context-Free | BCO | Convex | $\tilde{\mathcal{O}}\left(T^{3/4}/\varepsilon\right)$ | $\tilde{\mathcal{O}}\left(T^{3/4}\right)$ [16] |
| | | Convex + Smooth | $\tilde{\mathcal{O}}\left(T^{2/3}/\varepsilon\right)$ | $\tilde{\mathcal{O}}\left(T^{2/3}\right)$ [32] |
| | | S.C | $\tilde{\mathcal{O}}\left(T^{2/3}/\varepsilon\right)$ | $\tilde{\mathcal{O}}\left(T^{2/3}\right)$ [3] |
| | | S.C + Smooth | $\tilde{\mathcal{O}}\left(T^{1/2}/\varepsilon\right)$ | $\tilde{\mathcal{O}}\left(T^{1/2}\right)$ [18] |
| | MP-BCO | Convex | $\tilde{\mathcal{O}}\left(T^{1/2}/\varepsilon^2\right)$ | $\tilde{\mathcal{O}}\left(T^{1/2}\right)$ [3] |
| | | Strongly Convex | $\tilde{\mathcal{O}}\left(\log T/\varepsilon^2\right)$ | $\tilde{\mathcal{O}}\left(\log T\right)$ [3] |
| Context-Based | Contextual Linear Bandits | | $\tilde{\mathcal{O}}(T^{3/4}/\varepsilon)$ | $\tilde{\mathcal{O}}(T^{1/2})$ [1] |
| | Generalize Linear Bandits | | $\tilde{\mathcal{O}}(T^{3/4}/\varepsilon)$ | $\tilde{\mathcal{O}}(T^{1/2})$ [30] |

Table 1: Summary of our main results under $(\varepsilon, \delta)$-LDP, where $\tilde{\mathcal{O}}$ notation hides dependence over dimension $d$ and other poly-logarithmic factors. (S.C means Strongly Convex, MP means Multi-Point)

algorithm with $\tilde{\mathcal{O}}\left(T^{3/4}/\varepsilon\right)$ regret for convex loss and $\tilde{\mathcal{O}}\left(T^{2/3}/\varepsilon\right)$ regret for strongly convex loss, which nearly match current best non-private results under the same conditions [3, 16][1]. However, when loss functions are further smooth, current best non-private bounds for convex/strongly convex bandits are $\tilde{\mathcal{O}}\left(T^{2/3}\right)$ [32] and $\tilde{\mathcal{O}}\left(T^{1/2}\right)$ [18] respectively, and previous approaches [37, 5] seem hard to achieve such regret bounds in the same setting under privacy constraint (see Section 3.1 for more discussions). Besides BCO and its extension to multi-point feedback [3], context-free bandits also include other important cases, such as Multi-Armed Bandits (MAB), and there have been lots of algorithms designed for *differentially private* MAB [37, 31, 38, 39, 5, 33], either in stochastic or adversarial environment. As one can see, there are many different settings in context-free bandits learning, and existing differentially private algorithms are carefully designed for each one of them, which makes them relatively inconvenient to be used. Besides, their theoretical performance is analyzed separately and rather complicated. Some of them do not match corresponding non-private results.

Different with context-free bandits, usually there are certain contexts in real applications, such as user profile that contains user's features. Advanced bandit model uses these contexts explicitly to find the corresponding best action at each round, which is called contextual bandits. Two representatives are contextual linear bandits [29] and Generalized Linear Bandits [15]. Given benefits of contextual bandits, one may also wish to design corresponding private mechanisms. However, Shariff and Sheffet [35] proved that any *differentially private* contextual bandit algorithm would cause an $\Omega(T)$ regret bound. Hence, they considered a relaxed definition of DP called *joint differential privacy*, and proposed an algorithm based on LinUCB [1] with regret bound $\tilde{\mathcal{O}}\left(T^{1/2}/\varepsilon\right)$ [35] under $\varepsilon$-*joint differential privacy*.

Note all of previous study focus on *differential privacy* or its relaxed version. Compared with Differential Privacy, most of time Local Differential Privacy (LDP) [26, 11] is a much stronger and user-friendly standard of privacy and is more appealing in real applications [10], as LDP requires protecting each user's data before collection.

For context-free bandits, it is not hard to see algorithms with LDP guarantee protects DP automatically. However in contextual bandits, things become more delicate. These two definitions are not comparable as they have different interpretations about the output sequence, and traditional post-processing property cannot be used here to imply LDP is more rigorous than DP. In detail, DP regards predicted actions for contexts as the output sequence. Since optimal action varies from round to round in contextual bandits, it is not surprising there is a lower bound of linear regret in this case [35], as DP requires outputs to be nearly the same for any two neighboring datasets/contexts, which essentially contradicts with the goal of personalized prediction in contextual bandits. In contrast, LDP regards the collected information from users as "output sequence" and has no restriction on predicted actions, which is more reasonable as these actions are predicted on the local side based on local personal information and will not be released to public. Therefore, LDP seems like a more appropriate standard

for contextual bandits compared with DP, and maybe there is hope to bypass the lower bound proved for DP contextual bandits.

Given above discussions, a natural question arises: can we design simple and effective algorithms for bandits learning with LDP guarantee?

**Our Contributions:** In this work, we study both context-free bandits[2] and contextual bandits with LDP guarantee. Our contributions are summarized as follows: (see Table 1 for more details)

**(1)** We propose a simple reduction framework motivated by Agarwal and Singh [5] for a large class of context-free bandits learning problems with LDP guarantee, including BCO, MAB and Best Arm Identification (see Section 3.1 and Appendix[3] B). Equipped with different non-private algorithms, the utility of our framework can match corresponding best non-private performances, and these results are obtained through a unified and simple analysis;

**(2)** By modifying above framework slightly, we extend our algorithm to BCO with multi-point feedback [3], and design the *first* LDP multi-point BCO algorithm with nearly optimal guarantees;

**(3)** For contextual bandits including contextual linear bandits and more difficult generalized linear bandits, we propose algorithms with regret bounds $\tilde{\mathcal{O}}(T^{3/4}/\varepsilon)$ under $(\varepsilon, \delta)$-LDP , which are conjectured to be optimal. Note that these results show a fundamental difference between LDP and DP contextual bandits as discussed above.

All our results can be extended in parallel to $\varepsilon$-LDP if using Laplacian noise instead of Gaussian noise. Here, we only focus on $(\varepsilon, \delta)$-LDP.

**Comparison with Prior Work:** As mentioned earlier, for context-free bandits, nearly all of previous work focused on *differentially private* bandits learning, rather than stronger LDP guarantee. Only algorithms proposed in Tossou and Dimitrakakis [39] and Agarwal and Singh [5] for adversarial MAB can be converted to LDP version easily and obtain almost the same results. Though both their algorithms and ours are nearly the same in MAB, which is a very special case of bandits learning, our analysis is different, and we prove a new result for MAB with LDP guarantee as a side-product, which achieves nearly optimal regret bound under *both adversarial and stochastic environment simultaneously* (Appendix B.1). What's more, our results apply to more general bandits learning. For more comparison with Agarwal and Singh [5], see Section 3.1. Note, even in stronger LDP context-free bandits, our framework can achieve improved regret bounds for smooth BCO compared with previous results under weaker DP guarantee [37]. Besides, to the best of our knowledge, we give the first results for contextual bandits under LDP.

## 2 Preliminaries

**Notations:** $[p] = \{1, 2, \cdots, p\}$. $d$ is the dimension of decision space, and $e_i$ represents $i$-th basis vector. For a vector $x$ and a matrix $M$, define $\|x\|_M := \sqrt{x^\top M x}$. Given a set $\mathcal{W}$, we define the projection into this set as $\Pi_{\mathcal{W}}(\cdot)$.

Suppose the server collects certain information from each user with data domain $\mathcal{C}$. $\mathcal{C}$ can be the range of loss values in context-free bandits, or both contexts and losses/rewards in contextual bandits. Now we define LDP rigorously:

**Definition 1** (LDP). *A mechanism $Q : \mathcal{C} \to \mathcal{Z}$ is said to protect $(\varepsilon, \delta)$-LDP, if for any two data $x, x' \in \mathcal{C}$, and any (measurable) subset $U \subset \mathcal{Z}$, there is*

$$\Pr[Q(x) \in U] \leqslant e^\varepsilon \Pr[Q(x') \in U] + \delta$$

*In particular, if $\mathcal{Q}$ preserves $(\varepsilon, 0)$-LDP, we call it $\varepsilon$-LDP.*

Now, we introduce a basic mechanism in LDP literature – Gaussian Mechanism. Given any function $h : \mathcal{C} \to \mathbb{R}^d$. Define $\Delta := \max_{x, x' \in \mathcal{C}} \|h(x) - h(x')\|_2$, then Gaussian Mechanism is defined as $h(x) + Y$, where random vector $Y$ is sampled from Gaussian distribution $\mathcal{N}(0, \sigma^2 \mathrm{I}_d)$ with $\sigma = \frac{\Delta \sqrt{2 \ln(1.25/\delta)}}{\varepsilon}$. One can prove Gaussian Mechanism preserves $(\varepsilon, \delta)$-LDP [12].

**Algorithm 1:** One-Point Bandits Learning-LDP

---

**1** **Input**: non-private algorithm $\mathcal{A}$, privacy parameters $\varepsilon, \delta$

**2** **Initialize:** set $\sigma = \frac{2B\sqrt{2\ln(1.25/\delta)}}{\varepsilon}$

**3** **for** $t = 1, 2, \ldots$ **do**

**4** $\quad$ Server plays $x_t \in \mathcal{X}$ returned by $\mathcal{A}$;

**5** $\quad$ User $t$ suffers loss $f_t(x_t)$ and sends $f_t(x_t) + Z_t$ to $\mathcal{A}$ in the server, where $Z_t \sim \mathcal{N}(0, \sigma^2)$;

**6** $\quad$ $\mathcal{A}$ receives $f_t(x_t) + Z_t$ and calculates $x_{t+1}$

---

Next, we define the common strong convexity and smoothness for a function $f$.

**Definition 2.** *We say that a function $f : \mathcal{X} \to \mathbb{R}$ is $\mu$-strongly convex if there is: $f(x) - f(y) \leqslant \nabla f(x)^\top (x - y) - \frac{\mu}{2} \|x - y\|_2^2$. We say that a function $f : \mathcal{X} \to \mathbb{R}$ is $\beta$-smooth if it satisfies the following inequality: $\left| f(x) - f(y) - \nabla f(y)^\top (x - y) \right| \leqslant \frac{\beta}{2} \|x - y\|_2^2$*

## 3 Nearly Optimal Context-Free Bandits Learning with LDP Guarantee

In this section, we consider private context-free bandits learning with LDP guarantee, including bandits with one-point and multi-point feedback. As the following theorem shows, LDP is much stronger than DP in this setting (see Appendix A for the definition of DP in streaming setting and the proof), therefore it is more difficult to design algorithms under LDP with nearly optimal guarantee.

**Theorem 1.** *If an algorithm $\mathcal{A}$ protects $\varepsilon$-LDP, then any algorithm based on the output of $\mathcal{A}$ on a sequence of users guarantees $\varepsilon$-DP in streaming setting.*

### 3.1 Private Bandits Learning with One-Point Feedback

Bandits learning with one-point feedback includes several important cases, such as BCO, MAB, and Best Arm Identification (BAI). Generally speaking, we need to choose an action in the decision set at each round based on all previous information, then receive corresponding loss value of the action we choose. Most of time, our goal is to design an algorithm to minimize regret (it will be defined clearly later) compared with any fixed competitor.

Different with previous work [37, 31, 38, 39, 33], which designed delicate algorithms for different bandit learning problems under DP, here we propose a general framework to solve all of them within a unified analysis under stronger LDP. Our general private framework is shown in Algorithm 1, based on a pre-chosen non-private black-box bandits learning algorithm $\mathcal{A}$. Definitions of $\mathcal{X}, f_t$ and the choice of $\mathcal{A}$ in Algorithm 1 will be made clear in concrete settings below. Here we only assume all $f_t(x)$ are bounded by a constant $B$, i.e., $\forall x \in \mathcal{X}, t \in [T], |f_t(x)| \leqslant B$.

For private linear bandits learning, Agarwal and Singh [5] also propose a general reduction framework that can achieve nearly optimal regret. The key idea is to inject a linear perturbation $\langle n_t, x_t \rangle$ to the observed value $f_t(x_t)$ at each round, where $x_t$ is the current decision strategy and $n_t$ is fresh *noise vector* sampled from a predefined distribution. Because of the special form of linear loss, their approach actually protects data sequence in the *functional* sense, i.e., it is equivalent to disturbing original linear loss function $f_t(x)$ with noisy function $n_t^\top x$. However, this approach cannot protect privacy when loss functions are nonlinear, as injected noise depends on strategy $x_t$. Just consider $x_t = 0$, then it may leak the information of $f_t$ as values of different nonlinear functions can be different at point $x_t = 0$ and there is no noise at all if we use perturbation $\langle n_t, x_t \rangle$. Instead, our main idea is to inject fresh *noise variable* directly to the observed loss value at each round, which doesn't rely on $x_t$ any more. Intuitively, this approaches looks more natural as bandits learning algorithms only use the information of these observed loss values instead of loss functions.

Obviously, the LDP guarantee of Algorithm 1 is followed directly from basic Gaussian mechanism.

**Theorem 2.** *Algorithm 1 guarantees $(\varepsilon, \delta)$-LDP.*

To show the power of Algorithm 1, here we consider its main application, Bandits Convex Optimization. For another two concrete applications, MAB and BAI, see Appendix B for more details.

Besides, it also looks promising to extend the technique to pure exploration in combinatorial bandits (e.g., [21]).

In bandit convex optimization [20], $\mathcal{X}$ is a bounded convex constraint set. At each round, the server chooses a prediction $x_t$ based on previous collected information, then suffers and observers a loss value $f_t(x_t)$. The goal is to design an algorithm with low regret defined as $\max_{x \in \mathcal{X}} \mathbb{E}[\sum_{t=1}^{T} f_t(x_t) - f_t(x)]$. There are two different environments which generate underlying loss function sequence $\{f_t(x)|t \in [T]\}$. For adversarial BCO, there is no further assumption about $\{f_t(x)|t \in [T]\}$ and they are fixed functions given before games starts. For stochastic BCO [4], feedback $f_t(x_t)$ is generated as $f(x_t) + q_t$, where $f(x)$ is an unknown convex function and $\{q_t\}$ are independently and identically distributed noise sampled from a sub-Gaussian distribution $\mathcal{Q}$ with mean 0.

A critical ingredient in BCO is the gradient estimator constructed through the observed feedback. Besides convexity, when $f_t$ have additional properties like smoothness or strong convexity, usually we need to construct different gradient estimators and use different efficient non-private algorithms $\mathcal{A}$ to achieve better performance [16, 3, 32, 18]. Denote $u_t$ as a uniform random vector sampled from the unit sphere, then two representatives of gradient estimators are sphere sampling estimator $\frac{d}{\rho} f_t(x_t) u_t$ used in [16, 3] ($\rho$ is a parameter), and advanced ellipsoidal sampling estimator $d f_t(x_t) A_t^{-1} u_t$ which is the key part in [32, 18] to further improve the performance, where $A_t$ is the Hessian matrix induced by certain loss function with self-concordant barrier.

When it comes to private setting, Thakurta and Smith [37] designed a delicate *differentially private* algorithm with $\tilde{\mathcal{O}}\left(T^{3/4}/\varepsilon\right)$ and $\tilde{\mathcal{O}}\left(T^{2/3}/\varepsilon\right)$ guarantees for convex and strongly convex loss functions respectively, based on classical sphere sampling estimator and tree-based aggregation technique [14]. To achieve better bounds under additional smoothness assumption, it seems natural to combine their method with advanced ellipsoidal sampling estimator. However, this approach doesn't work even under DP guarantee, let alone LDP guarantee. In detail, to protect privacy, usually we need to add noise proportional to the range of information we use. For classical sphere sampling estimator, it is bounded by $dB/\rho$. However, for the advanced ellipsoidal sampling estimator, the spectral norm of inverse Hessian of self-concordant barrier (i.e., $A_t^{-1}$) can be unbounded, which makes it hard to protect privacy. Besides, tree-based aggregation techniques fail in LDP setting.

Instead of adding noise to the accumulated estimated gradient like Thakurta and Smith [37], our general reduction Algorithm 1 injects noise directly to the loss value that is already bounded. Based on the critical observation that the regret defined for original loss functions $\{f_t(x)|t \in [T]\}$ equals to the regret defined for virtual loss functions $\{f_t(x) + Z_t|t \in [T]\}$ in expectation, we avoid complex analysis which is based on a connection with non-private solutions [37], and obtain the utility of our private algorithm through the guarantee of non-private algorithm $\mathcal{A}$ directly as the following shows:

**Theorem 3.** *Suppose non-private algorithm $\mathcal{A}$ achieves regret $B \cdot \mathrm{Reg}_{\mathcal{A}}^T$ for BCO, where $B$ is the range of loss function. We have the following guarantee for Algorithm 1: for any $x \in \mathcal{X}$, there is*

$$\mathbb{E}\left[\sum_{t=1}^{T} f_t(x_t) - f_t(x)\right] \leqslant \tilde{\mathcal{O}}\left(\frac{B \ln(T/\delta)}{\varepsilon} \cdot \mathrm{Reg}_{\mathcal{A}}^T\right) \tag{1}$$

*where expectation is taken over the randomness of non-private algorithm $\mathcal{A}$ and all injected noise.*[4]

With above theorem, by plugging different non-private optimal algorithms under variant cases, we obtain corresponding regret bounds with LDP guarantee:

**Corollary 4.** *When loss functions are convex and $\beta$-smooth, Algorithm 1 achieves $\tilde{\mathcal{O}}(T^{2/3}/\varepsilon)$ regret by setting $\mathcal{A}$ as Algorithm 1 in [32]. When loss functions are $\mu$-strongly convex and $\beta$-smooth, Algorithm 1 achieves $\tilde{\mathcal{O}}(\sqrt{T}/\varepsilon)$ regret by setting $\mathcal{A}$ as Algorithm 1 in [18]. For private Stochastic BCO, using Algorithm 2 in [4] as the black-box algorithm will achieve $\tilde{\mathcal{O}}(\sqrt{T}/\varepsilon)$ regret.*

Note this result improves previous result [37] in three aspects. First, our Algorithm 1 guarantees stronger LDP rather than DP. Second, it achieves better regret bounds when loss functions are further smooth, and matches corresponding non-private results. Third, our algorithm is easy to be implemented, admits a unified analysis, and also obtains new results in stochastic BCO.

**Algorithm 2:** Two-Point Feedback Private Bandit Convex Optimization via Black-box Reduction

---

1 **Input**: set $\mathcal{A}$ as Algorithm 4 (in Appendix C) with parameters $\eta, \rho, \xi$, privacy parameters $\varepsilon, \delta$

2 **Initialize:** set $\sigma = \frac{2G\sqrt{2\ln(1.25/\delta)}}{\varepsilon}, \eta = \frac{1}{\sqrt{T}}, \rho = \frac{\log T}{T}, \xi = \frac{\rho}{r}$

3 **for** $t = 1, 2, \dots$ **do**

4     Server plays $x_{t,1}, x_{t,2} \in \mathcal{X}$ received from $\mathcal{A}$

5     User suffers $f_t(x_{t,1}), f_t(x_{t,2})$ and passes $f_t(x_{t,1}) - f_t(x_{t,2}) + n_t^\top(x_{t,1} - x_{t,2})$ to $\mathcal{A}$ in the server, where $n_t \sim \mathcal{N}(0, \sigma^2 \mathrm{I}_d)$

---

### 3.2 Private Bandits Convex Optimization with Multi-Point Feedback

Now we consider BCO with Multi-Point Feedback. Different with one-point bandit feedback setting, where we can only query one point at each round, now we can query multiple points. This is natural in many applications, such as in personalized recommendation, we can recommend multiple items to each user and receive their feedback. Suppose we are permitted to query $K$ points per round (denote them as $x_{t,1}, \dots, x_{t,K}$ at round $t$), then we observe $f_t(x_{t,1}), \dots, f_t(x_{t,K})$. Suppose decision set $\mathcal{X}$ satisfies $r\mathbb{B} \subset \mathcal{X} \subset R\mathbb{B}$ like in Agarwal et al. [3], where $\mathbb{B}$ is the unit ball in $\mathbb{R}^d$. The expected regret is defined as

$$\mathbb{E}\left[\frac{1}{K}\sum_{t=1}^{T}\sum_{k=1}^{K} f_t(x_{t,k})\right] - \min_{x \in \mathcal{X}} \mathbb{E}\left[\sum_{t=1}^{T} f_t(x)\right] \tag{2}$$

where $\{f_t(x)\}$ are $G$-Lipschitz convex functions, and expectation is taken over the randomness of algorithm.

With the relaxation of amount about queries, there is a significant difference about regret bound of BCO between one-point feedback and $K$-point feedback for $K \geqslant 2$ [3]. In detail, the minimax regret for general BCO with one-point feedback is in order $\tilde{\mathcal{O}}(\sqrt{T})$ (even for strongly convex and smooth losses [34]), whereas one can design algorithms for BCO under multi-point feedback with $\mathcal{O}(\sqrt{T})$ regret for convex loss and $\mathcal{O}(\log T)$ regret for strongly convex loss, just like full information online convex optimization. As there is not much difference between $K = 2$ and $K > 2$, so we focus on $K = 2$ in this paper. An optimal non-private algorithm can be found in [3] and is given as Algorithm 4 in Appendix C for completion, which will be used as our black-box algorithm later.

For private version of this problem, note our previous reduction framework no longer fits in this new setting, mainly because of multiple feedback. If we add the same noise $Z_t$ to observed values $f_t(x_{t,1}), f_t(x_{t,2})$, then it cannot guarantee privacy. If we use different noise $Z_{t,1}, Z_{t,2}$ to perturb observed values respectively, though it protects privacy, previous utility analysis fails.

Based on the non-private algorithm, we design a slightly modified reduction framework that resembles the approach in Agarwal and Singh [5] but for Multi-Point BCO, as shown in Algorithm 2. The key observation is that now we play two pretty close points $x_{t,1}, x_{t,2}$ at each round, and critical information we use about user $t$ is only the difference $f_t(x_{t,1}) - f_t(x_{t,2})$ of two observed values. Note $x_{t,1} - x_{t,2} = 2\rho u_t$ (see Algorithm 4 in Appendix C), which implies we can add noise $n_t^\top(x_{t,1} - x_{t,2})$ to $f_t(x_{t,1}) - f_t(x_{t,2})$ to protect its privacy. As $f_t(x)$ is $G$-Lipschitz, hence $|f_t(x_{t,1}) - f_t(x_{t,2})| \leqslant 2\rho G \|u_t\|_2$ and adding Gaussian noise with standard deviation $\sigma = \frac{2G\sqrt{2\ln(1.25/\delta)}}{\varepsilon}$ is enough to protect privacy as $\|u_t\|_2 = 1$.

**Theorem 5.** *Algorithm 2 guarantees $(\varepsilon, \delta)$-LDP.*

For utility analysis of Algorithm 2, as now the noise depends on strategies $x_{t,1}, x_{t,2}$ at round $t$, hence both output and regret in terms of original loss functions $\{f_t(x)|t \in [T]\}$ are the same as output and regret in terms of virtual loss functions $\{f_t(x) + n_t^\top x | t \in [T]\}$ in expectation. Therefore we can obtain the utility of our private Algorithm 2 through the guarantee of non-private algorithm $\mathcal{A}$:

**Theorem 6.** *For any $x \in \mathcal{X}$, Algorithm 2 guarantees*

$$\mathbb{E}\left[\frac{1}{2}\sum_{t=1}^{T}(f_t(x_{t,1}) + f_t(x_{t,2})) - f_t(x)\right] \leqslant \tilde{\mathcal{O}}\left(\frac{d^3\sqrt{T}}{\varepsilon^2}\right) \tag{3}$$

*If $\{f_t\}$ are further $\mu$ strongly convex, set $\eta = \frac{1}{\mu t}, \rho = \frac{\log T}{T}, \xi = \frac{\rho}{r}$, then for any $x \in \mathcal{X}$, we have*

$$\mathbb{E}\left[\frac{1}{2}\sum_{t=1}^{T}(f_t(x_{t,1}) + f_t(x_{t,2})) - f_t(x)\right] \leqslant \tilde{\mathcal{O}}\left(\frac{d^3 \log T}{\mu \varepsilon^2}\right) \tag{4}$$

From above results, one can see there is also a significant difference about regret bounds between BCO and Multi-Point BCO under LDP setting, which is exactly the same as non-private settings.

## 4 Contextual Bandits Learning with LDP Guarantee

In this section, we turn our attention to more practical contextual bandits learning. At each round $t$, the learner needs to choose an action $x_t \in \mathcal{X}_t$ in the local side, where $\mathcal{X}_t$ contains the personal information and features about underlying arms. Then the user generates a reward which is assumed to be $y_t = g(x_t^\top \theta^*) + \eta_t$, where $\theta^*$ is an unknown true parameter in the domain $\mathcal{W}$, $g : \mathbb{R} \to \mathbb{R}$ is a known function, and $\eta_t$ is a random noise in $[-1, 1]$ with mean 0 [5]. If we know $\theta^*$, $x_{t,*} := \arg\max_{x \in \mathcal{X}_t} g(x^\top \theta^*)$ is apparently the optimal choice at round $t$. For an algorithm $\mathcal{A}$, we define its regret over $T$ rounds as $\text{Reg}_T^{\mathcal{A}} := \sum_{t=1}^{T} g(x_{t,*}^\top \theta^*) - g(x_t^\top \theta^*)$, where $\{x_t, t \in [T]\}$ is the output of $\mathcal{A}$. We omit the superscript $\mathcal{A}$ when it is clear. There are two critical parts in contextual bandits. One is to estimate $\theta^*$, and corresponding estimated parameter is used to find best action for exploitation. Another one is to construct certain term for the purpose of exploration, since we are in the environment of partial feedback. Throughout this section, we assume both $\{\mathcal{X}_t\}$ and $\mathcal{W}$ are bounded by a $d$-dimensional $L_2$ ball with radius 1 for simplicity.

Compared with private context-free bandits, private contextual bandits learning is more difficult, not only because of relatively complicated setting, but we need to protect more information including both contexts and rewards, which causes additional difficulty in the analysis of regret. As a warm-up, we show how to design algorithm with LDP guarantee for contextual linear bandits, which resembles a recent work [35] but under a relaxed version of DP. Next, we propose a more complicated algorithm for generalized linear bandits with LDP guarantee.

### 4.1 Warm-Up: LDP Contextual Linear Bandits

In contextual linear bandits, mapping $g$ is an identity, or equivalently, the reward generated by user $t$ for action $x_t$ is $y_t = x_t^\top \theta^* + \eta_t$. To estimate $\theta^*$, the straightforward method is to use linear regression based on collected data. Combined with classic principal for exploration, optimism in the face of uncertainty, it leads to LinUCB [1], which is nearly optimal for contextual linear bandits. To protect privacy, it's not surprising that we adopt the same technique as LDP linear regression [36], i.e. injecting noise to $x_t x_t^\top$ and $y_t x_t$ collected from user $t$. However, the injected noise have influence not only over the parameter estimation, but also for further exploration part, due to more complex bandit model, thus we need to set parameters more carefully. See Algorithm 5 in Appendix D.

Now, we state the theoretical guarantee of Algorithm 5.

**Theorem 7.** *Algorithm 5 guarantees $(\varepsilon, \delta)$-LDP.*

**Theorem 8.** *With probability at least $1 - \alpha$, the regret of Algorithm 5 satisfies the following bound:*

$$\text{Reg}_T \leqslant \tilde{\mathcal{O}}\left(\sqrt{\log\frac{1}{\delta}\log\frac{1}{\alpha}}\frac{(dT)^{3/4}}{\varepsilon}\right) \tag{5}$$

Given the $\Omega(T)$ lower bound for DP contextual linear bandits [35], Theorem 8 implies a fundamental difference between LDP and DP in contextual bandit learning, which also verifies that LDP is a more appropriate standard about privacy for contextual bandits as discussed in the introduction. One may think we can still prove DP based on LDP guarantee and post-processing property. Recall post-processing property holds only for the output of a DP algorithm which doesn't use private data any more. However, in our algorithms for LDP contextual bandits, though we can use post-processing property to prove estimation sequence $\{\tilde{\theta}_t\}$ satisfies DP, it doesn't imply the output action sequence $\{x_t\}$ satisfies DP, as these actions are made in the local side which use private local data.

**Algorithm 3:** Generalized Linear Bandits with LDP

---

**1** **Input:** privacy parameters $\varepsilon, \delta$, failure probability $\alpha$

**2** **Initialize:** $\tilde{V}_0 = 0_{d \times d}, \tilde{u}_0 = 0_d, \tilde{\theta}_0 = \hat{\theta}_1 = 0_d, \zeta = \Theta(1/\sqrt{T}), \sigma = 6\sqrt{2\ln(3.75/\delta)}/\varepsilon$

**3** **Notations:** $\Upsilon_t = \sigma\sqrt{t}(4\sqrt{d} + 2\ln(2T/\alpha)), c_t = 2\Upsilon_t, \beta_t^2 = \tilde{\mathcal{O}}(\frac{C\sigma}{\mu}\sqrt{dt})$

**4** **for** $t = 1, 2, \dots$ **do**

**5**     **For the local user** $t$:

**6**     Receive information $\tilde{V}_{t-1}, \tilde{\theta}_{t-1}, \hat{\theta}_t$ from the server

**7**     Play action $x_t = \mathrm{argmax}_{x \in \mathcal{D}_t} \left\langle \tilde{\theta}_{t-1}, x \right\rangle + \beta_{t-1} \|x\|_{\tilde{V}_{t-1}^{-1}}$

**8**     Observe reward $y_t = g(x_t^\top \theta^*) + \eta_t$, set $z_t = x_t^\top \hat{\theta}_t$.

**9**     Send $x_t x_t^\top + B_t, z_t x_t + \xi_t, \nabla \ell_t(\hat{\theta}_t) + r_t$ to the server, where

      $\ell_t(\theta) = \ell(x_t^\top \theta, y_t), B_t(i,j) \overset{i.i.d}{\sim} \mathcal{N}(0, \sigma^2), \forall i \leqslant j$, and

      $B(j,i) = B(i,j), \xi_t \sim \mathcal{N}(0_d, \sigma^2 \mathrm{I}_{d \times d}), r_t \sim \mathcal{N}(0_d, C^2 \sigma^2 \mathrm{I}_{d \times d})$

**10**     **For the server:**

**11**     Update $\bar{V}_t = \bar{V}_{t-1} + x_t x_t^\top + B_t, \tilde{u}_t = \tilde{u}_{t-1} + z_t x_t + \xi_t \, \tilde{\theta}_t = \tilde{V}_t^{-1} \tilde{u}_t$, where $\tilde{V}_t = \bar{V}_t + c_t \mathrm{I}_{d \times d}$

      $\hat{\theta}_{t+1} = \Pi_{\mathcal{W}} \left( \hat{\theta}_t - \zeta(\nabla \ell_t(\hat{\theta}_t) + r_t) \right)$

---

### 4.2 LDP Generalized Linear Bandits

In generalized linear bandits, mapping $g$ can be regarded as the inverse link function of exponential family model. Here we suppose function $g$ is $G$-Lipschitz, continuously differentiable on $[-1, 1]$, $|g(a)| \leqslant C$, and $\inf_{a \in (-1,1)} g'(a) = \mu > 0$, which implies $g$ is strictly increasing. These assumptions are common either in real applications or previous work [30, 25]. We also define corresponding negative log-likelihood function $\ell(a, b) := -ab + m(a)$, where $m(\cdot)$ is the integral of function $g$. As a concrete example, if reward $y$ is a Bernoulli random variable, then the form of $g$ is $g(a) = (1 + \exp(-a))^{-1}, m(a) = \log(1 + \exp(a)), \ell(a, b) = \log(1 + \exp(-a(2b-1))), b \in \{0, 1\}$, and noise $\eta$ is $1 - g(a)$ with probability $g(a)$ and $-g(a)$ otherwise.

Note the non-linearity of $g$ makes things much more complicated either from the view of bandits learning or privacy preservation. The counterpart of Contextual Linear Bandits is linear regression, the locally private version of which is relatively easy and well-studied. However, the counterpart of Generalized Linear Bandit is Empirical Risk Minimization (ERM) with respect to generalized linear loss, and the optimal approach of parameter estimation for GLM bandit is to solve ERM at each round [30]. Different with linear regression, for learning ERM with LDP guarantee, in general there is no efficient private algorithm that can achieve optimal performance in the *non-interactive* environment [36, 42, 41], let alone calculating an accurate parameter estimation needed in our problem. Therefore, it seems hard to learn generalized linear bandit under LDP guarantee.

Luckily, we can make full use of the *interactive* environment in bandit problems. In detail, we build our private mechanism based on GLOC framework proposed in [25]. Compared with previous nearly optimal approach [30], GLOC framework enjoys much better time efficiency, which calculates estimator $\theta$ in an online fashion instead of solving ERM at each round. Its main idea is to maintain a rough estimation for unknown parameter $\theta^*$ through an adversarial online learning algorithm and use it to relabel current reward, and then solve the corresponding linear regression for a refined estimator. To achieve optimal $\tilde{\mathcal{O}}(\sqrt{T})$ regret, the online learning algorithm is set as Online Newton Step [19].

Though the original goal of GLOC framework proposed in Jun et al. [25] is to improve time efficiency, the update form of estimated parameter for unknown $\theta^*$ shares the same form of linear regression, therefore we can use nearly the same technique as in previous subsection to protect LDP, which avoids solving complex ERM with LDP guarantee. Besides, since internal online learning algorithm also utilizes users' data, we also need to guarantee its privacy. Different with Jun et al. [25] which adopts Online Newton Step, we choose basic noisy Online Gradient Descent as our online black-box algorithm. See Algorithm 3 for the full implementation. For clarity, we just write the LDP Online Gradient Descent explicitly in Line 11.

Though Algorithm 3 is based on the framework proposed by Jun et al. [25], we want to emphasize that both finding the right approach and proving the rigorous guarantee are non-trivial because of stringent LDP constraint. Following theorems give both the privacy guarantee and utility bound of our Algorithm 3 for generalized linear bandits.

**Theorem 9.** *Algorithm 3 guarantees $(\varepsilon, \delta)$-LDP.*

**Theorem 10.** *With probability at least $1 - \alpha$, the regret of Algorithm 3 satisfies the following bound:*

$$\mathrm{Reg}_T \leqslant \tilde{\mathcal{O}} \left( \sqrt{\log \frac{1}{\delta} \log \frac{1}{\alpha} \log \frac{T}{d}} \frac{(dT)^{3/4}}{\varepsilon} \right) \tag{6}$$

Note that both our upper bounds (5) and (6) are in order $\tilde{\mathcal{O}}\left(T^{3/4}\right)$, which differ from common $\mathcal{O}(\sqrt{T})$ regret bound in corresponding non-private settings. We conjecture this order is nearly the best one can achieve in LDP setting, mainly because we need to protect more information, i.e., both contexts and corresponding rewards. See Appendix G for more discussions and intuitions.

## 5 Conclusions

In this paper, we propose a simple black-box reduction framework that can solve a large class of context-free bandits learning problems with LDP guarantee in a unified way, including BCO, MAB, Best Arm Identification. We also extend the reduction framework to BCO with Multi-Point Feedback. This black-box reduction mainly has three advantages compared with previous work. First it guarantees a more rigorous LDP guarantee instead of DP. Second, this framework gives us a unified analysis for all above private bandit learning problems instead of analyzing each of them separately, and it easily improves previous best results or obtains new results for some problems, as well as matching corresponding non-private optimal bounds. Third, such a black-box reduction is more attractive in real applications, as we only need to modify the input to black-box algorithms. Besides, we also propose new algorithms for more practical contextual bandits with LDP guarantee, including contextual linear bandits and generalized linear bandits. Our algorithms can achieve $\tilde{\mathcal{O}}(T^{3/4})$ regret bound, which is conjectured to be nearly optimal. We leave the rigorous proof of this lower bound as an interesting open problem.

#### Broader Impact
This work is mostly theoretical, with no negative outcomes. (Contextual) bandits learning has been widely used in real applications, which heavily relies on user's data that may contain personal private information. To protect user's privacy, we adopt the appealing solid notion of privacy – Local Differential Privacy (LDP) that can protect each user's data before collection, and design (contextual) bandit algorithms under the guarantee of LDP. Our algorithms can be easily used in real applications, such as recommendation, advertising, to protect data privacy and ensure the utility of private algorithms simultaneously, which will befit everyone in the world.

## Acknowledgments and Disclosure of Funding

This work was supported by National Key R&D Program of China (2018YFB1402600), Key-Area Research and Development Program of Guangdong Province (No. 2019B121204008), Beijing Academy of Artificial Intelligence, and in part by the Zhongguancun Haihua Institute for Frontier Information Technology.

## Footnotes

[1] Kwai Inc.

[2] School of Mathematical Sciences, Peking University

[3] Haihua Institute for Frontier Information Technology

[4] Huawei Noah's Ark Lab

[5] Key Laboratory of Machine Perception, MOE, School of EECS, Peking University

[6] Center for Data Science, Peking University

[1]Though Bubeck et al. [9] designed a polynomial time algorithm for general BCO with $\tilde{\mathcal{O}}(T^{1/2})$ regret, it is far from practical, so we don't consider its result in this paper, but of course we can plug that algorithm into our framework to obtain optimal $\tilde{\mathcal{O}}(T^{1/2}/\varepsilon)$ bound for general private BCO.

[2]Note that adaptive adversary is ambiguous in bandits setting [6], so we only consider oblivious adversary throughout the paper.

[3]Appendix could be found in the full version [43].

[4]Actually, if using the high probability guarantee of black-box algorithm $\mathcal{A}$, we can also obtain corresponding high probability guarantee of our Algorithm 1. See Appendix E for more details, and the same argument there can be extended to results in section 3.2 as well.

[5] It's not hard to relax this constraint to a sub-Gaussian noise.

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
