[Supplementary Material · supp.pdf]

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

## Appendix

## A   Differential Privacy under streaming setting

Differential Privacy [13] is original proposed for off-line setting. Later, Dwork et al. [14] and Jain et al. [22] consider DP in streaming setting. In streaming setting, at each round $t$, the server predicts $x_t \in \mathcal{X}$ for user $t$ whose personal data is represented as $h_t \in \mathcal{H}$ (for example, his or her feature, label, or preference etc.). Then the server requires some information $z_t \in \mathcal{Z}$ from user $t$ ($z_t$ may depend on $x_t$ and $h_t$) to update the model for next prediction. Note DP allows collecting true data (i.e. $z_t = h_t$) and is defined in terms of the output sequence $\{x_t\}$, while LDP doesn't allow collecting true data and is defined in terms of the collected information $z_t$. Here we adopt the definition given in Jain et al. [22] for DP in streaming setting:

**Definition 3** (Differential Privacy). *Let $F = \langle h_1, h_2, \ldots, h_T \rangle$ be a sequence of information which domain is $\mathcal{H}^{1:T}$. Let $\mathcal{A}(F) = Y$, where $Y = \langle y_1, y_2, \ldots, y_T \rangle \in \mathcal{Y}^{1:T}$ be $T$ outputs of the randomized algorithm $\mathcal{A}$. $\mathcal{A}$ is said to preserve $(\varepsilon, \delta)$-differential privacy, if for any two information sequences $F, F'$ that differ in at most one entry, and for any subset $S^{1:T} \subset \mathcal{Y}^{1:T}$, it holds that*

$$\Pr(\mathcal{A}(F) \in S^{1:T}) \le \Pr(\mathcal{A}(F') \in S^{1:T})e^{\varepsilon} + \delta.$$

*In particular, if $\mathcal{A}$ preserves $(\varepsilon, 0)$-differential privacy, we say $\mathcal{A}$ is $\varepsilon$-differentially private.*

Now, we prove Theorem 1:

*Proof of Theorem 1.* Suppose algorithm $\mathcal{A} : \mathcal{H} \to \mathcal{Z}$ protects $\varepsilon$-LDP, that is for any $h, h' \in \mathcal{H}, U \subset \mathcal{Z}$, we have

$$\Pr(\mathcal{A}(h) \in U) \leqslant e^{\varepsilon} \times \Pr(\mathcal{A}(h') \in U)$$

Denote $\mathcal{G}$ as arbitrary online/bandits algorithm received the output of $\mathcal{A}$ on user sequence, i.e. $\{z_t = \mathcal{A}(h_t|x_t)|t \in [T]\}$. Now we prove $\mathcal{G}$ protects $\varepsilon$-DP, i.e. for any $S^{1:T} \subset \mathcal{X}^{1:T}$ and neighboring sequence $F = \{h_t|t \in [T]\}, F' = \{h'_t|t \in [T]\}$ that only differ in one entry, we have the following inequality:

$$\Pr(\mathcal{G}(\mathcal{A}(F)) \in S^{1:T}) \leqslant e^{\varepsilon} \times \Pr(\mathcal{G}(\mathcal{A}(F')) \in S^{1:T})$$

Without loss of generality, we assume $F$ and $F'$ differ in the $t$-th entry. Since $\mathcal{G}$ only operates on $\{z_t|t \in [T]\}$, according to the Post-Processing property of DP [12], we only need to prove $\{z_t|t \in [T]\}$ satisfies $\varepsilon$-DP. Denote $\{z'_t|t \in [T]\}$ as the neighboring information sequence of $\mathcal{A}$ operated on $F'$, then for arbitrary $U^{1:T} \subset \mathcal{Z}^{1:T}$ we have

$$\frac{\Pr(z_{1:T} \in U^{1:T})}{\Pr(z'_{1:T} \in U^{1:T})} \tag{7}$$

$$=\frac{\Pr(z_{1:t-1} \in U^{1:t-1}) \times \Pr(z_t \in U^t|z_{1:t-1} \in U^{1:t-1}) \times \Pr(z_{t+1:T} \in U^{t+1:T}|z_{1:t} \in U^{1:t})}{\Pr(z'_{1:t-1} \in U^{1:t-1}) \times \Pr(z'_t \in U^t|z'_{1:t-1} \in U^{1:t-1}) \times \Pr(z'_{t+1:T} \in U^{t+1:T}|z'_{1:t} \in U^{1:t})} \tag{8}$$

$$=\frac{\Pr(z_t \in U^t|z_{1:t-1} \in U^{1:t-1})}{\Pr(z'_t \in U^t|z'_{1:t-1} \in U^{1:t-1})} \tag{9}$$

$$=\frac{\Pr(z_t \in U^t|x_t \in \mathcal{G}(U^{1:t-1}))}{\Pr(z'_t \in U^t|x'_t \in \mathcal{G}(U^{1:t-1}))} \tag{10}$$

$$\leqslant e^{\varepsilon} \tag{11}$$

where the second equation is because two data sequence only differ at round $t$, and $\mathcal{G}$ operates on the sequence of $z$. Thus we prove the theorem. □

## B   Another Two Applications for Bandits Learning with One-point Feedback

### B.1   Private Multi-Armed Bandits

MAB is a special case of BCO, in which decision set $\mathcal{X} = \{e_i|i \in [d]\}$, and loss function $f_t(x)$ is actually a linear function, i.e. $f_t(x) = \ell_t^\top x$, where $\ell_t \in [0, 1]^d$. In the adversarial setting, sequence $\{\ell_t\}$ is chosen arbitrarily before game starts. In stochastic setting, for each arm $k$, $\{\ell_t(k)\}$ are

independently sampled from underlying unknown distribution $\mathcal{V}_k$ with support over interval $[0, 1]$. Denote $\mu_k$ as the expected loss of arm $k$. Without loss of generality, assume $\mu_1 > \mu_2 > \cdots > \mu_d$ and define $\Delta_i := \mu_i - \mu_d$. It is well-known the optimal regret are $\mathcal{O}(\sqrt{dT})$ and $\mathcal{O}(\sum_{i:\Delta_i > 0} \frac{\log T}{\Delta_i})$ for adversarial MAB and stochastic MAB respectively [8]. However, especially in real applications, usually we don't know whether we are in adversarial or stochastic environment in advance. Until recently, Zimmert and Seldin [44] proposed a single algorithm achieving the optimal performance for both adversarial and stochastic world *without* any prior information about the environment.

For differentially private MAB, all of previous work consider either stochastic loss *or* adversarial loss [31, 38, 39, 5]. While here, we hope to handle both scenarios simultaneously like in non-private case but with LDP guarantee. Not surprisingly, by plugging the non-private optimal algorithm [44] in our black-box, we obtain corresponding private version which achieves the best of both adversarial and stochastic worlds:

**Theorem 11.** *By choosing non-private black-box algorithm $\mathcal{A}$ in Algorithm 1 as* TSALLIS-INF *in Zimmert and Seldin [44] and setting $\sigma$ as in Theorem 2 with $B = 0.5$,*

- *in the adversarial setting, we have*

$$\max_{x \in \mathcal{X}} \mathbb{E}\left[ \sum_{t=1}^{T} \ell_t(x_t) - \ell_t(x) \right] \leqslant \tilde{\mathcal{O}}\left( \frac{\sqrt{T}}{\varepsilon} \right) \tag{12}$$

- *in the stochastic setting, we have*

$$\max_{x \in \mathcal{X}} \mathbb{E}\left[ \sum_{t=1}^{T} \ell_t(x_t) - \ell_t(x) \right] \leqslant \tilde{\mathcal{O}}\left( \sum_{i:\Delta_i > 0} \frac{\log T}{\Delta_i \varepsilon^2} \log \frac{1}{\delta} \right) \tag{13}$$

Note above results not only nearly match corresponding non-private lower bounds [8] regardless of privacy parameters, but also lower bounds under LDP restriction [7]. Besides, we can also use many other MAB algorithms as our black-box candidates such as KL-UCB [17] Stochastic MAB, which will then obtain more delicate bound under LDP.

## B.2 Private Best Arm Identification

Different with Stochastic MAB, in which one has to balance between *Exploration* and *Exploitation*, Best Arm Identification (BAI) problem only focuses on the *Exploration*, that is finding the best arm among all arms. Here we use same notations as Subsection B.1. There are mainly two settings in BAI: *fixed confidence setting* and *fixed budget setting*. In this part, we only consider *fixed confidence setting*: given any confidence parameter $\gamma$, design an algorithm which outputs the best arm with probability at least $1 - \gamma$ using as fewest samples as possible [23, 27]. It's not hard to see our method can be generalized to fixed budget setting as well.

For private BAI, though algorithms in Mishra and Thakurta [31] and Sajed and Sheffet [33] are designed for stochastic MAB, they can also used for differentially private BAI. However, these algorithms only achieve sub-optimal guarantee, let alone stronger LDP. While here, we want to protect LDP and achieve nearly optimal sample complexity. Again, using the same observation as Subsection B.1 and given any non-private BAI algorithm $\mathcal{A}$, our Algorithm 1 has the following guarantee:

**Theorem 12.** *Given any confidence parameter $\gamma$, suppose non-private BAI algorithm $\mathcal{A}$ achieves sample complexity $\mathrm{SA}(\mathcal{A}, \sigma_0^2, \gamma)$, where $\sigma_0^2$ is the variance proxy parameter of underlying unknown sub-Gaussian distributions $\{\mathcal{V}_k | k \in [d]\}$. Set $\sigma$ as in Theorem 2 with $B = 0.5$, then the sample complexity of Private BAI Algorithm 1 is $\mathrm{SA}(\mathcal{A}, \frac{1}{4} + \sigma^2, \gamma)$.*

*Specifically, if we choose non-private BAI algorithm $\mathcal{A}$ as lil'UCB in Jamieson et al. [24], then the sample complexity of Algorithm 1 is in order $\mathcal{O}\left( \sum_{k \neq 1} \frac{\ln\left( (\ln 1/\Delta_k^2)/\gamma \right)}{\varepsilon^2 \Delta_k^2} \ln \frac{1}{\delta} \right)$.*

## C  Non-private Algorithm for Bandits Learning with Two-points Feedback

For completeness, we present the non-private algorithm proposed in Agarwal et al. [3] for Bandits Convex Optimization with two-point feedback. See Algorithm 4.

---

**Algorithm 4:** Expected Gradient Descent with two queries per round [3]

---

1  **Input**: Learning rate $\eta$, exploration parameter $\rho$ and shrinkage coefficient $\xi$

2  Set $y_1 = 0$

3  **for** $t = 1, 2, \ldots$ **do**

4     Pick a unit vector $u_t$ uniformly at random

5     Play $x_{t,1} := y_t + \rho u_t, x_{t,2} := y_t - \rho u_t$, and observe $f_t(x_{t,1}), f_t(x_{t,2})$

6     Set $\tilde{g}_t = \frac{d}{2\rho}\left(f_t(x_{t,1}) - f_t(x_{t,2})\right) u_t$

7     update $y_{t+1} = \prod_{(1-\xi)\mathcal{X}}(y_t - \eta \tilde{g}_t)$, where $\prod_{\mathcal{X}}$ represents projection to the set $\mathcal{X}$

---

---

**Algorithm 5:** Contextual Linear Bandits with LDP

---

1  **Input:** privacy parameters $\varepsilon, \delta$, failure probability $\alpha$.

2  **Initialize:** $\tilde{V}_0 = 0_{d\times d}, \tilde{u}_0 = 0_d, \tilde{\theta}_0 = 0_d, \sigma = 6\sqrt{2\ln(2.5/\delta)}/\varepsilon$.

3  **Notations:** $\Upsilon_t = \sigma\sqrt{t}(4\sqrt{d} + 2\ln(2T/\alpha)), c_t = 2\Upsilon_t$,

    $\beta_t = 2\sigma\sqrt{d\ln T} + \left(\sqrt{3\Upsilon_t} + \sigma\sqrt{\frac{dt}{\Upsilon_t}}\right)d\ln T$.

4  **for** $t = 1, 2, \ldots, T$ **do**

5     **For the local user** $t$**:**

6     Receive information $\tilde{V}_{t-1}, \tilde{\theta}_{t-1}$ from the server.

7     Play action $x_t = \arg\max_{x \in \mathcal{D}_t}\left\langle \tilde{\theta}_{t-1}, x\right\rangle + \beta_t \|x\|_{(\tilde{V}_{t-1}+c_{t-1}\mathrm{I})^{-1}}$

8     Observe reward $y_t = \langle x_t, \theta^*\rangle + \eta_t$

9     Send $x_t x_t^\top + B_t, y_t x_t + \xi_t$ to the server, where $B_t(i,j) \overset{i.i.d}{\sim} \mathcal{N}(0, \sigma^2), \forall i \leqslant j$, and
    $B(j,i) = B(i,j), \xi_t \sim \mathcal{N}(0_d, \sigma^2\mathrm{I}_{d\times d})$.

10     **For the server:** update

11     $\tilde{V}_t = \tilde{V}_{t-1} + x_t x_t^\top + B_t, \tilde{u}_t = \tilde{u}_{t-1} + y_t x_t + \xi_t$

12     $\tilde{\theta}_t = \left(\tilde{V}_t + c_t\mathrm{I}_{d\times d}\right)^{-1}\tilde{u}_t$

---

## D   Contextual Linear Bandits with LDP

See Algorithm 5 above.

## E   Omitted Proofs in Section 3

*Proof of Theorem 2.* Since for any $x \in \mathcal{X}, t \in [T], |f_t(x)| \leqslant B$, which means the sensitivity of information sent from the user is at most $2B$, thus $(\varepsilon, \delta)$-LDP property of Algorithm 1 follows directly from the Gaussian mechanism.

$\square$

*Proof of Theorem 3.* Note all noise are independently sampled, hence we can fix $Z_1, \ldots Z_T$ in advance. Define pseudo loss $\tilde{f}_t(x) = f_t(x) + Z_t$. According to the tail bound of Gaussian variable, there is

$$\Pr\left[|Z_t| > \sigma\sqrt{2\ln 2T^2}\right] \leqslant \frac{1}{T^2} \tag{14}$$

By union bound, we have

$$\Pr\left[\exists t \in [T], |Z_t| > \sigma\sqrt{2\ln 2T^2}\right] \leqslant \frac{1}{T} \tag{15}$$

Define the event $F := \{\exists t \in [T] : |Z_t| > \sigma\sqrt{2\ln 2T^2}\}$, then there is $\Pr[F] \leqslant \frac{1}{T}$.

Once fixed $Z_1, \ldots, Z_T$, the output of running Algorithm 1 over loss sequence $\{f_t | t \in [T]\}$ is the same as the output of running non-private algorithm $\mathcal{A}$ over pseudo loss sequence $\{\tilde{f}_t | t \in [T]\}$.

On one hand, we have

$$\mathbb{E}\left[\sum_{t=1}^{T}\tilde{f}_t(x_t)-\tilde{f}_t(x)\right] \leqslant \mathbb{E}\left[\sum_{t=1}^{T}\tilde{f}_t(x_t)-\tilde{f}_t(x)|\bar{F}\right]+\Pr[F]\times\mathbb{E}\left[\sum_{t=1}^{T}\tilde{f}_t(x_t)-\tilde{f}_t(x)|F\right] \quad (16)$$

$$\leqslant \mathbb{E}\left[\sum_{t=1}^{T}\tilde{f}_t(x_t)-\tilde{f}_t(x)|\bar{F}\right]+2B \quad (17)$$

$$\leqslant (B+\sigma\sqrt{2\ln(2T^2)})\cdot\mathrm{Reg}_{\mathcal{A}}^T+2B \quad (18)$$

On the other hand, according to our definition of $\tilde{f}_t(x)$, there is always

$$\sum_{t=1}^{T}\tilde{f}_t(x_t)-\tilde{f}_t(x)=\sum_{t=1}^{T}f_t(x_t)-f_t(x) \quad (19)$$

Combine above equations, we obtain the conclusion.

For the high probability version, suppose black-box algorithm $\mathcal{A}$ guarantees that: for any loss sequence $\{\tilde{f}_t(x)\}$ with loss range $\tilde{B}$, with probability at least $1-\kappa$ (over the internal randomness of $\mathcal{A}$), there is

$$\forall x\in\mathcal{X}, \sum_t\tilde{f}_t(x_t)-\sum_t\tilde{f}_t(x)\leqslant\tilde{B}\cdot\mathrm{Reg}_{\mathcal{A}}^T \quad (20)$$

According to union bound and above discussion, we know: $\tilde{B}=B+\sigma\sqrt{2\ln(2T^2)}$, and with probability at least $1-\kappa-\frac{1}{T}$, there is

$$\forall x\in\mathcal{X}, \sum_t f_t(x_t)-\sum_t f_t(x)\leqslant\tilde{\mathcal{O}}\left(\frac{B\ln(T/\delta)}{\varepsilon}\cdot\mathrm{Reg}_{\mathcal{A}}^T\right) \quad (21)$$

$\square$

*Proof of Corollary 4.* The guarantee for (strongly) convex and smooth bandit optimization is straightforward by plugging corresponding non-private guarantees in Saha and Tewari [32], Hazan and Levy [18]. For Stochastic BCO, since our algorithm is equivalent to the case of running any stochastic BCO algorithm over new noise distribution $\mathcal{Q}\bigotimes\mathcal{N}(0,\sigma^2)$, where $\bigotimes$ represents the convolution between two distributions, we can use the guarantee for stochastic BCO in Agarwal et al. [4]. $\square$

*Proof of Theorem 11.* In adversarial setting, using Theorem 3 obtains the regret bound. Now we prove the regret bound in stochastic setting. Note for any $k\in[d]$, as the support of original distribution $\mathcal{V}_k$ is over $[0,1]$, it is a sub-Gaussian distribution with variance proxy $\frac{1}{4}$. Define pseudo distribution $\tilde{\mathcal{V}}_k=\mathcal{V}_k\bigotimes\mathcal{N}(0,\sigma^2)$, where $\bigotimes$ represents the convolution between two distributions. Obviously, the output of Algorithm 1 over distributions $\{\mathcal{V}_k|k\in[K]\}$ is the same as the output of non-private algorithm $\mathcal{A}$ over distributions $\{\tilde{\mathcal{V}}_k|k\in[K]\}$. As $\tilde{\mathcal{V}}_k$ is now a sub-Gaussian with variance proxy $\frac{1}{4}+\sigma^2$, hence it's not hard to obtain the conclusion according to the guarantee of $\mathcal{A}$. $\square$

*Proof of Theorem 12.* Just use Theorem 2 in the paper Jamieson and Nowak [23] with new sub-Gaussian parameter $\frac{1}{4}+\sigma^2$ $\square$

*Proof of Theorem 5.* Since $|f_t(x_{t,1})-f_t(x_{t,2})|\leqslant 2\rho G\|u_t\|_2=2\rho G$ and $n_t^\top(x_{t,1}-x_{t,2})=2\rho n_t^\top u_t$ which obeys $\mathcal{N}(0,4\rho^2\sigma^2)$, the privacy guarantee then follows according to Gaussian mechanism.

$\square$

*Proof of Theorem 6.* Note all noise vectors are independently sampled, hence we can fix $n_1,\ldots,n_T$ in advance. Define pseudo loss $\tilde{f}_t(x)=f_t(x)+n_t^\top x$. For any $\{u_t|t\in[T]\}$ in the unit sphere, according to the tail bound of Gaussian variable, there is

$$\Pr\left[|n_t^\top u_t|>\sigma\sqrt{2\ln 2T^2}\right]\leqslant\frac{1}{T^2} \quad (22)$$

By union bound, we have

$$\Pr\left[\exists t \in [T], |n_t^\top u_t| > \sigma\sqrt{2\ln 2T^2}\right] \leqslant \frac{1}{T} \tag{23}$$

Define the event $F := \{\exists t \in [T] : |n_t^\top u_t| > \sigma\sqrt{2\ln 2T^2}\}$, then there is $\Pr[F] \leqslant \frac{1}{T}$.

Once fixed $n_1, \ldots, n_T$, the output of running Algorithm 2 over loss sequence $\{f_t | t \in [T]\}$ is the same as the output of running non-private Algorithm 4 over pseudo loss sequence $\{\tilde{f}_t | t \in [T]\}$.

On one hand, we have

$$\mathbb{E}\left[\frac{1}{2}\sum_{t=1}^{T}\left(\tilde{f}_t(x_{t,1}) + \tilde{f}_t(x_{t,2})\right) - \tilde{f}_t(x)\right] \tag{24}$$

$$\leqslant \mathbb{E}\left[\frac{1}{2}\sum_{t=1}^{T}\left(\tilde{f}_t(x_{t,1}) + \tilde{f}_t(x_{t,2})\right) - \tilde{f}_t(x)|\bar{F}\right] + \Pr[F] \times \mathbb{E}\left[\frac{1}{2}\sum_{t=1}^{T}\left(\tilde{f}_t(x_{t,1}) + \tilde{f}_t(x_{t,2})\right) - \tilde{f}_t(x)|F\right] \tag{25}$$

$$\leqslant \mathbb{E}\left[\frac{1}{2}\sum_{t=1}^{T}\left(\tilde{f}_t(x_{t,1}) + \tilde{f}_t(x_{t,2})\right) - \tilde{f}_t(x)|\bar{F}\right] + 2B \tag{26}$$

$$\leqslant \mathrm{Reg}(\mathcal{A}, G + \sigma\sqrt{d}) + 2B \tag{27}$$

where $\mathrm{Reg}(\mathcal{A}, G + \sigma\sqrt{d})$ represents the regret bound of non-private Algorithm 4 for loss functions with Lipschitz constant $G + \sigma\sqrt{d}$. On the other hand, there is

$$\mathbb{E}\left[\sum_{t=1}^{T}\tilde{f}_t(x_t) - \tilde{f}_t(x)\right] = \mathbb{E}\left[\sum_{t=1}^{T}f_t(x_t) - f_t(x)\right] \tag{28}$$

Combine above equations with the guarantee of non-private Algorithm 4 in Agarwal et al. [3], we obtain the conclusion. □

## F  Omitted Proofs in Section 4

*Proof of Theorem 7.* Since $\|x_t\| \leqslant 1, y_t \in [-2, 2]$ according to our assumption, the privacy guarantee then follows directly from the Gaussian Mechanism, as both the matrix and vector sent to the server satisfy $(\varepsilon/3, \delta/2)$-LDP and $(2\varepsilon/3, \delta/2)$-LDP respectively. □

*Proof of Theorem 8.* Note our private matrix $\tilde{V}_t$ is an unbiased estimation of true matrix $\sum_{s=1}^{t} x_s x_s^\top$ with noise $H_t := \sum_{s=1}^{t} B_s$, where its upper triangular entry obeys the distribution $\mathcal{N}(0, t\sigma^2)$. Similarly, $\tilde{u}_t$ is an unbiased estimation of true vector $\sum_{s=1}^{t} y_s x_s$ with noise $h_t := \sum_{s=1}^{t} \xi_s$, where $h_t \sim \mathcal{N}(0_d, t\sigma^2 \mathrm{I}_{d \times d})$. According to the concentration inequality [40], we know $\|H_t\|_2 \leqslant \sigma\sqrt{t}(4\sqrt{d} + 2\ln(2T/\alpha)) = \Upsilon_t$ with probability at least $1 - \alpha/2T$, thus all the eigenvalues of $H_t + c_t \mathrm{I}_{d \times d}$ are in the range $[\Upsilon_t, 3\Upsilon_t]$ with high probability. Besides, we have $\|h_t\|_{(H_t + c_t \mathrm{I}_{d \times d})^{-1}} \leqslant \sqrt{\Upsilon_t^{-1}}\|h_t\|_2$, and $\|h_t\|_2 \leqslant \sigma\sqrt{dt}$ with high probability. Now, using Proposition 4, Proposition 11 and Theorem 5 in paper [35] with our noise, we obtain the conclusion. □

*Proof of Theorem 9.* Since $\|x_t\| \leqslant 1, |z_t| \leqslant 1$, and loss function $\ell_t$ is $C$-Lipschitz, the privacy guarantee follows directly from the Gaussian Mechanism, as the matrix, vector, and gradient of any user sent to the server satisfy $(\varepsilon/3, \delta/3)$-LDP respectively. □

*Proof of Theorem 10.* Define instantaneous regret $r_t : g(x_{t,*}^\top \theta^*) - g(x_t^\top \theta^*)$, then there is $r_t \leqslant G(x_{t,*}^\top \theta^* - x_t^\top \theta^*)$. Besides

$$x_t^\top \theta^* + 2\beta_{t-1}\|x_t\|_{\tilde{V}_{t-1}^{-1}} \geqslant x_t^\top \theta^* + \left\|\tilde{\theta}_{t-1} - \theta^*\right\|_{\tilde{V}_{t-1}}\|x_t\|_{\tilde{V}_{t-1}^{-1}} + \beta_{t-1}\|x_t\|_{\tilde{V}_{t-1}^{-1}}$$

$$\geqslant x_t^\top \tilde{\theta}_{t-1} + \beta_{t-1}\|x_t\|_{\tilde{V}_{t-1}^{-1}}$$

$$\geqslant x_{t,*}^\top \tilde{\theta}_{t-1} + \beta_{t-1} \left\| x_{t,*} \right\|_{\tilde{V}_{t-1}^{-1}}$$

$$\geqslant x_{t,*}^\top \theta^*$$

where the second and the forth inequality is because of our Confidence Ellipsoid Lemma 2. Thus we have $r_t \leqslant 2G\beta_{t-1} \left\| x_t \right\|_{\tilde{V}_{t-1}^{-1}}$.

Next using common technique in contextual bandits to bound $\sum_t \left\| x_t \right\|_{\tilde{V}_{t-1}^{-1}}$ [35, 25], we have $\sum_t r_t \leqslant G\beta_T \sqrt{dT \log T}$, which finishes the proof. $\qquad\square$

**Lemma 1** (Regret of LDP-OGD). *For any convex loss sequence $\{\ell_t(\theta)|t \in [T]\}$ with Lipschitz constant $C$, and for $\forall \theta \in \Theta$, with probability at least $1 - \alpha_1$, we have the following bound*

$$\sum_{t=1}^T \ell_t(\hat{\theta}_t) - \ell_t(\theta) \leqslant \mathcal{O}\left( C\sigma \sqrt{dT \ln \frac{T}{\alpha_1}} \right) \tag{29}$$

*where $\{\hat{\theta}_t | t \in [T]\}$ are outputs of noisy OGD like step 13 in Algorithm 3, and the randomness is over noise $\{r_t | t \in [T]\}$.*

*Proof.* Condition on the event $\mathcal{E} = \{\forall t \in [T], \left\| r_t \right\|_2 \leqslant \sqrt{d}C\sigma\}$ (which happens with high probability) and according to the guarantee of On-line Gradient Descent [20], there is $\sum_t \ell_t(\hat{\theta}_t) + r_t^\top \hat{\theta}_t - (\ell_t(\theta) + r_t^\top \theta) \leqslant \mathcal{O}(C\sigma\sqrt{dT})$. Next, using martingale concentration, we know $\left\| \sum_t r_t^\top \hat{\theta}_t \right\|_2 \leqslant \sigma\sqrt{dT}$ and $\left\| \sum_t r_t^\top \theta \right\|_2 \leqslant C\sigma\sqrt{dT}$ with high probability. Combining above three inequalities, we obtain the conclusion. $\qquad\square$

**Lemma 2** (Confidence Ellipsoid). *In terms of Algorithm 3, with probability at least $1 - \alpha_2$, we have the following bound*

$$\forall t, \quad \left\| \tilde{\theta}_t - \theta^* \right\|_{\tilde{V}_t}^2 \leqslant \tilde{\mathcal{O}}\left( \frac{C\sigma}{\mu} \sqrt{dT \ln \frac{T}{\alpha_2}} \right) \tag{30}$$

*where $\{\tilde{\theta}_t, \tilde{V}_t | t \in [T]\}$ are outputs of Algorithm 3, and the randomness is over the injected noise as well as underlying environment.*

*Proof.* Since $\inf_{a \in (-1,1)} g'(a) = \mu > 0$, it implies loss function $\ell(a, b)$ is $\mu$-strongly convex in terms of the first argument, thus

$$\sum_{s=1}^t \ell_s(\hat{\theta}_s) - \ell_s(\theta^*) = \sum_{s=1}^t \ell(x_s^\top \hat{\theta}_s, y_s) - \ell(x_s^\top \theta^*, y_s) \tag{31}$$

$$\geqslant \sum_{s=1}^t \ell'(x_s^\top \theta^*, y_s)(x_s^\top \hat{\theta}_s - x_s^\top \theta^*) + \frac{\mu}{2}(x_s^\top \hat{\theta}_s - x_s^\top \theta^*)^2 \tag{32}$$

$$= \sum_{s=1}^t (-y_s + g(x_s^\top \theta^*))(x_s^\top \hat{\theta}_s - x_s^\top \theta^*) + \frac{\mu}{2}(x_s^\top \hat{\theta}_s - x_s^\top \theta^*)^2 \tag{33}$$

$$= \sum_{s=1}^t -\eta_s(x_s^\top \hat{\theta}_s - x_s^\top \theta^*) + \frac{\mu}{2}(x_s^\top \hat{\theta}_s - x_s^\top \theta^*)^2 \tag{34}$$

Then according to Lemma 1 above, with probability at least $1 - \alpha_1$, there is

$$\frac{\mu}{2} \sum_{s=1}^t (x_s^\top \hat{\theta}_s - x_s^\top \theta^*)^2 \leqslant \mathcal{O}\left( C\sigma \sqrt{dt \ln \frac{T}{\alpha_1}} \right) + \sum_{s=1}^t \eta_s(x_s^\top \hat{\theta}_s - x_s^\top \theta^*) \tag{35}$$

Using Corollary 8 in paper [2], with probability at least $1 - \alpha_3$ (over the randomness of noise $\{\eta_t\}$), for all $t$, there is

$$\sum_{s=1}^t \eta_s(x_s^\top \hat{\theta}_s - x_s^\top \theta^*) \leqslant \sqrt{\left( 2 + 2\sum_{s=1}^t (x_s^\top(\hat{\theta}_s - \theta^*))^2 \right) \cdot \ln\left( \frac{1}{\alpha_3} \sqrt{1 + \sum_{s=1}^t (x_s^\top(\hat{\theta}_s - \theta^*))^2} \right)} \tag{36}$$

Combine above two inequalities, and solve the right hand side using Lemma 2 in paper [25], then with probability $1 - \alpha_1 - \alpha_3$, we have

$$\forall t, \quad \sum_{s=1}^{t}(x_s^\top(\hat\theta_s - \theta^*))^2 \leqslant \tilde{\mathcal{O}}\left(\frac{C\sigma}{\mu}\sqrt{dt\ln\frac{T}{\alpha_1}\ln\frac{T}{\alpha_3}}\right) \tag{37}$$

Denote $X_t \in \mathbb{R}^{t\times d}$ as the design matrix consisting of $x_1, \ldots, x_t$, $Z_t = [z_1; z_2; \ldots; z_t] \in \mathbb{R}^t$, $\bar{B}_t = \sum_{s=1}^{t} B_s, \bar{\xi}_t = \sum_{s=1}^{t}\xi_s$. Note

$$\sum_{s=1}^{t}(x_s^\top(\hat\theta_s - \theta^*))^2 \tag{38}$$

$$= \|\theta^*\|_{X_t^\top X_t}^2 - 2Z_t^\top X_t\theta^* + \|Z_t\|_2^2 \tag{39}$$

$$= \|\theta^*\|_{\tilde{V}_t}^2 - 2\tilde{u}_t^\top\theta^* + \|Z_t\|_2^2 - \|\theta^*\|_{\bar{B}_t+\tilde{V}_0}^2 + 2\bar{\xi}_t^\top\theta^* \tag{40}$$

$$= \left\|\theta^* - \tilde\theta_t\right\|_{\tilde{V}_t}^2 - \left\|\tilde\theta_t\right\|_{\tilde{V}_t}^2 + \|Z_t\|_2^2 - \|\theta^*\|_{\bar{B}_t+\tilde{V}_0}^2 + 2\bar{\xi}_t^\top\theta^* \tag{41}$$

$$= \left\|\theta^* - \tilde\theta_t\right\|_{\tilde{V}_t}^2 + \left\|X_t\tilde\theta_t - Z_t\right\|_2^2 - \left\|\tilde\theta_t\right\|_{X_t^\top X_t}^2 + 2\tilde\theta_t^\top X_t^\top Z_t - \left\|\tilde\theta_t\right\|_{\tilde{V}_t}^2 - \|\theta^*\|_{\bar{B}_t+\tilde{V}_0}^2 + 2\bar{\xi}_t^\top\theta^* \tag{42}$$

$$= \left\|\theta^* - \tilde\theta_t\right\|_{\tilde{V}_t}^2 + \left\|X_t\tilde\theta_t - Z_t\right\|_2^2 - \left\|\tilde\theta_t\right\|_{X_t^\top X_t}^2 + 2\tilde\theta_t^\top\tilde{u}_t - \left\|\tilde\theta_t\right\|_{\tilde{V}_t}^2 - \|\theta^*\|_{\bar{B}_t+c_t\mathrm{I}}^2 + 2\bar{\xi}_t^\top(\theta^* - \tilde\theta_t) \tag{43}$$

$$= \left\|\theta^* - \tilde\theta_t\right\|_{\tilde{V}_t}^2 + \left\|X_t\tilde\theta_t - Z_t\right\|_2^2 - \left\|\tilde\theta_t\right\|_{X_t^\top X_t}^2 + 2\left\|\tilde\theta_t\right\|_{\tilde{V}_t}^2 - \left\|\tilde\theta_t\right\|_{\tilde{V}_t}^2 - \|\theta^*\|_{\bar{B}_t+c_t\mathrm{I}}^2 + 2\bar{\xi}_t^\top(\theta^* - \tilde\theta_t) \tag{44}$$

$$= \left\|\theta^* - \tilde\theta_t\right\|_{\tilde{V}_t}^2 + \left\|X_t\tilde\theta_t - Z_t\right\|_2^2 + \left\|\tilde\theta_t\right\|_{\bar{B}_t+c_t\mathrm{I}}^2 - \|\theta^*\|_{\bar{B}_t+c_t\mathrm{I}}^2 + 2\bar{\xi}_t^\top(\theta^* - \tilde\theta_t) \tag{45}$$

$$\geqslant \left\|\theta^* - \tilde\theta_t\right\|_{\tilde{V}_t}^2 - \|\theta^*\|_{\bar{B}_t+c_t\mathrm{I}}^2 - 2\left\|\bar{\xi}_t\right\|_2 - 2\bar{\xi}_t^\top\tilde\theta_t \tag{46}$$

Combine above inequalities, there is

$$\left\|\theta^* - \tilde\theta_t\right\|_{\tilde{V}_t}^2 \leqslant \tilde{\mathcal{O}}\left(\frac{C\sigma}{\mu}\sqrt{dt\ln\frac{T}{\alpha_1}\ln\frac{T}{\alpha_3}}\right) + \|\theta^*\|_{\bar{B}_t+c_t\mathrm{I}}^2 + 2\left\|\bar{\xi}_t\right\|_2 + 2\bar{\xi}_t^\top\tilde\theta_t \tag{47}$$

On the other hand, with probability at least $1 - \alpha_4$, there is

$$\|\theta^*\|_{\bar{B}_t+c_t\mathrm{I}}^2 \leqslant \tilde{\mathcal{O}}(\Upsilon_t) = \tilde{\mathcal{O}}(\sigma\sqrt{dt}) \tag{48}$$

$$\left\|\bar{\xi}_t\right\|_2 \leqslant \tilde{\mathcal{O}}(\sigma\sqrt{dt}) \tag{49}$$

and

$$\bar{\xi}_t^\top\tilde\theta_t \leqslant \bar{\xi}_t^\top\tilde{V}_t^{-1}(X_t^\top Z_t + \bar{\xi}_t) \tag{50}$$

$$\leqslant \bar{\xi}_t^\top\tilde{V}_t^{-1}X_t^\top Z_t + \tilde{\mathcal{O}}(\sigma\sqrt{dt}) \tag{51}$$

$$\leqslant \tilde{\mathcal{O}}(\sigma\sqrt{dt}) \tag{52}$$

where the last inequality is because $\tilde{V}_t^{-1}X_t^\top Z_t$ is the solution of regularized least square loss function $J(\theta) := \|X_t\theta - Z_t\|_2^2 + \|\theta\|_{c_t\mathrm{I}+\bar{B}_t}^2$. Since $J(\theta^*) \leqslant \tilde{\mathcal{O}}(\sigma\sqrt{dt})$, and $\Upsilon_t\mathrm{I} \leqslant c_t\mathrm{I} + \bar{B}_t \leqslant 3\Upsilon_t\mathrm{I}$ holds with high probability, there is $\left\|\tilde{V}_t^{-1}X_t^\top Z_t\right\| \leqslant \tilde{O}(1)$, otherwise it cannot be the solution of $J(\theta)$.

Thus, with probability at least $1 - \alpha_1 - \alpha_3$, we have

$$\left\|\tilde\theta_t - \theta^*\right\|_{\tilde{V}_t}^2 \leqslant \mathcal{O}\left(\frac{C\sigma}{\mu}\sqrt{dt\ln\frac{T}{\alpha_1}\ln\frac{T}{\alpha_3}}\right) \tag{53}$$

Taking a union bound over all $T$ rounds and choose appropriate $\alpha_1, \alpha_3$ we then finish the proof. $\quad\square$

# G   Discussion about Lower Bound in LDP Contextual Bandits

Either for contextual linear bandits or more complex generalized linear bandits, both of our algorithms with LDP guarantee can only achieve $\tilde{\mathcal{O}}(T^{3/4})$ regret, contrasted with optimal $\mathcal{O}(T^{1/2})$ regret in non-private case [30], as well nearly optimal $\tilde{\mathcal{O}}(T^{1/2})$ regret for MAB with LDP guarantee. The critical difference is that we need to protect more information in contextual bandits. If we regard MAB as a special case of contextual bandits, decision set $\mathcal{X}_t$ then becomes $\{e_i | i \in [d]\}$. Privacy of contexts means we need to protect $(e_{I_t}, r_t)$ sent from user $t$ to the server at round $t$, where $I_t$ is the chosen arm and $r_t$ is the reward of user $t$. Recall in Section 3.1, we only protect $r_t$. Denote $\theta_t$ as the estimation of underlying $\theta^*$ at round $t$, and define $M_t := \sum_{\tau=1}^{t} e_{I_\tau} e_{I_\tau}^\top$. Roughly speaking, in almost all analysis of stochastic MAB, the regret bound depends on $\tilde{\mathcal{O}}(\sqrt{T} \|\theta_T - \theta^*\|_{M_T})$, and $\|\theta_T - \theta^*\|_{M_T}$ is nearly a constant in either non-private setting or our MAB example in Appendix B.1. However in the setting of this section, on one hand, for those sub-optimal arms $i$, the algorithm won't play it too much, and its estimation error $|\theta_T(i) - \theta^*(i)|$ is roughly in constant order. On the other hand, since we still need to protect $e_{I_t}$ at each round, which will lead to an estimation error of $M_T$ in order $\sqrt{T}$. Therefore $\|\theta_T - \theta^*\|_{M_T}$ is roughly in order $\tilde{\mathcal{O}}(T^{1/4})$ under LDP setting, which leads to the final $\tilde{\mathcal{O}}(T^{3/4})$ regret. Though this special case looks a little strange, it shows an inherent difficulty in contextual bandits learning with LDP guarantee, and we conjecture that $\Omega(T^{3/4})$ is exactly the lower bound in this case.