[Reviews · NeurIPS 2020]

Review 1

Summary and Contributions: The paper proposes to consider local differential privacy, rather than differential privacy, for bandit learning. For context-free bandits, LDP is stronger than DP, yet the paper provides similar guarantees with a unified algorithm. For contextual bandits, LDP is not comparable to DP, but LDP allows the authors to circumvent a linear regret lower bound for DP and provide sublinear regret LDP algorithms.

Strengths: The paper seems to match or improve on existing results for private bandits using a unified framework. I generally find such results are helpful for improving our conceptual understanding. The paper does a good job of discussing related work. However, I am not an expert in this area so I could not fully judge the context of related work or novelty/correctness of the proofs.

Weaknesses: There were some grammar errors that distracted from the clarity of the writing. I also felt the paper could be better sign-posted so that it is clear where each section is going, where to find different results, overall flow, etc. That being said, the paper was clear enough for me to understand it; I would mainly recommend the authors spend some more time carefully proofreading and thinking about how to sign-post / organize the work.

Correctness: The claims seem reasonable and plausible, but I did not assess them carefully.

Clarity: See above.

Relation to Prior Work: See above.

Reproducibility: Yes

Additional Feedback: To assess the claim that the results match or improve prior work, it would help to report the best existing results for regular DP analogous to Table 1. Can you please do so in the author response? A clearer comparison to existing results would potentially increase my score. In the broader impacts statement, I found the argument that there could be no negative impacts because the work is theoretical to be unconvincing.


Review 2

Summary and Contributions: This paper studies locally differentially private (LDP) bandits learning. First, they propose simple black-box reduction frameworks that can solve a large family of context-free bandits learning problems with LDP guarantee. Based on the frameworks, they improve previous best results for private bandits learning with one-point feedback, such as private Bandits Convex Optimization, and obtain the first result for Bandits Convex Optimization (BCO) with multi-point feedback under LDP. Further, they extend the (ε, δ)-LDP algorithm to Generalized Linear Bandits, which enjoys a sub-linear regret and is conjectured to be nearly optimal. Their result also shows a fundamental difference between LDP and DP contextual bandits learning.

Strengths: They consider the Local Differential Privacy (LDP), not the differential privacy in many previous work, which is a much stronger and user-friendly standard of privacy and is more appealing in real applications. They also show private algorithms with good regret bounds.

Weaknesses: For multi-point setting, in Algorithm 2, the noise depends on the Lipschitz parameter G, so this works for the loss function for G-Lipschitz, this condition is not shown in the Table 1.

Correctness: Authors's feedback helps me to understand my following questions. The proofs in appendix are not checked step by step. For contextual bandit, this work shows a sub-linear regret for LDP, but there exists a linear regret for DP. Intuitively, LDP is a stronger version of DP. Authors explain that this is because they consider the collected information, rather than the predicted action, as the output. But to my knowledge, the prediction can be obtained from the collected information, usually from the summation of all rounds. Since the collected information is LDP in each round, the collected information from all rounds will be DP, and the prediction is also DP by the post-process. Therefore, it will have a contradiction to the lower bound. I’ll be willing for some more (intuitive) explanations from authors to convince me that the above thought has mistake and the result in paper is correct. For multi-point setting, in Algorithm 6 and Table 1, the regrets for MP are not consistent, one is over epsilon squared, the other is over epsilon. There are some other confusions of the proofs. For instance, in the proof of Theorem 6, Eq(27), authors say the where Reg(A, G + σ √d) represents the regret bound of non-private Algorithm 4. Why this regret can be gotten from Eq(26) could be explained, since it is not trivial.

Clarity: The paper is well written, but there are also some parts could be improved. The definition of their learning problems (such as BCO,MAB,etc.) and the regret benchmark could be introduced in preliminaries, to be friendly for reading. What’s more, they say LDP and DP are not comparable for contextual bandit learning because that they has different interpretations about the output and LDP regards the collected information as output, which could be specified which information it is. Without knowing what this information is, this result looks confusing for me. In Algorithm 1, Row 4 and Row 6 have redundancy.

Relation to Prior Work: Authors show the comparison between their work to previous works.

Reproducibility: Yes

Additional Feedback:


Review 3

Summary and Contributions: The paper considers the problem of private bandit optimization under both contextual and context-free setting. The main contributions are the following: 1. For the single-point feedback model in the context-free setting, the paper provides a black-box method that guarantees eps, delta differential privacy by perturbing the loss feedback and provides regret bounds that are a factor of log(T/delta)/eps higher than the non-private counterpart. 2. For the multi-point feedback model in the context-free setting, the paper provides algorithms that achieve \sqrt{T}/eps and (\log T)/eps regret bounds when the loss functions are convex and strongly convex respectively. 3. Under the contextual bandit setting, the paper considers a framework where only loss feedbacks instead of contexts are sent back to the server (discussed later). The authors provide algorithms that achieve regret bounds of T^{3/4}/\espunder this framework under generalized linear models. // After authors' feedback: I have read the authors' feedback. I will keep my score. I hope the authors can make the distinction between their setting (the LDP requirement is on the updates) and the regular LDP setting where the privacy requirement is on the access to the data in the final version if it gets accepted.

Strengths: For the single-point feedback model in the context-free setting, the proposed black-box method can be combined with any non-private bandit optimization algorithm with at most a factor of log(T/delta)/eps increase in the regret bound. Moreover, the proposed method guarantee a stronger notion of Local differential privacy. The paper also proposes the first algorithm to address multi-point feedback model in the context-free setting under LDP constraints. The bound also matches the non-private counterpart in terms of T.

Weaknesses: 1. It is worth pointing out that the randomized purterbation steps in the algorithms for the context-free setting are similar to the one proposed in [Agarwal and Singh 2017]. 2. This question is possibly out of the scope of the paper. Is the dependence on eps in the reduction optimal? In [Agarwal Singh 2017] and [Tossou and Dimitrakakis 2015], they obtain bounds that in the form of (non-private regret + 1/eps), i.e., the price of the loss is additive. It would be nice if the authors can comment on this or tightness of the bounds for the context-free setting. 3. For the contextual setting, it is not accurate to say that the algorithm guarantees a stronger notion of LDP compared to DP since the setting is different from the one considered in [Shariff and Sheffet 2018]. In the model considered in this paper, the recommendation sets are stored locally and the recommendation is also computed locally while only the feedbacks and models are exchanged between the server and the users. This model requires computation overhead at the users. I agree this model is valuable and I like the results. However, I think it would be nice if the authors can make the comparisons clear.

Correctness: I haven't read all the proofs in the appendix. But the results seem correct to me.

Clarity: The writing of the paper is clear and organized.

Relation to Prior Work: To the best of my knowledge, the paper addresses previous adequantly.

Reproducibility: Yes

Additional Feedback:


Review 4

Summary and Contributions: The paper considers bandit algorithms protecting privacy of users providing information to the bandit algorithms. As a measure of privacy protection, the differential privacy (DP) is considered well, and several bandit algorithms satisfying conditions on DP are presented in the previous studies. In contrast to the previous studies, the paper considers local differential privacy (LDP), which is a stronger notion than the DP. For several fundamental variants of the bandit problems, the paper presents algorithms satisfying the LDP condition and analyses their regret bounds. The bounds match the regret bounds of non-private bandit algorithms in many cases, and improves the known bounds achieved by several algorithms satisfying the DP conditions.

Strengths: - The paper introduces the concept of LDP into the bandit algorithm. The LDP is stronger than the DP because it protects the privacy of information before collecting them, and thus it is more favorable. Moreover, the paper shows that every algorithm satisfying the LDP condition also satisfies the DP condition except for the contextual bandits. - The analysis of regret bounds are non-trivial. To satisfy the LDP condition, it is usual to add noise to rewards given from the chosen arms, and the proposed algorithms also take this approach. However, how much the noise affects the regrets of algorithms. The paper analyzes this point rigorously.

Weaknesses: The paper has no experiments. It is more interesting to see the trade-off between the privacy and the performance of algorithms.

Correctness: I do not check details of the claims although they seem reliable. The analysis highly relies on the previous studies.

Clarity: It is better to explain the frameworks of DP and LDP more kindly. For example, the LDP condition is explained in Section 2, but it was not clear to me what is the mechanism Q in the bandit problems. In Theorem 1, "any algorithm based on the output of A" is not a clear expression.

Relation to Prior Work: The paper explains previous studies on private and non-private bandit algorithms well.

Reproducibility: Yes

Additional Feedback: I read the authors' feedback and my opinion hasn't changed.

[Author Response · NeurIPS 2020]

We thank all reviewers for their valuable comments, and will improve our writing (like correcting grammar error,
reorganizing main results, explaining relations with previous work, introducing various definitions more clear) in further
revision. Below we respond to several raised issues. Here, the reference citation number is the same as main submission.

**Reviewer 1:**
— "report the best existing results for regular DP analogous to Table 1 (in the paper)":

Please see right table for the comparison between our results in LDP setting and the best existing results in DP setting. As one can see, in smooth BCO setting (either convex or strongly convex), we achieve improved regret bounds compared with DP counterparts (note in context-free setting, LDP bandit learning is strictly harder than DP bandit learning). For multi-point BCO, as far as we know, there is no prior study in this setting. In contextual bandit setting, LDP and DP are not comparable, and [34] proved a lower bound of linear regret for DP contextual linear bandit, which is a special case of generalized linear bandits. Given this lower bound, our results show there is a fundamental difference between LDP and DP contextual bandits learning.

| Type | Problem | | LDP (Ours) | DP (Existing) |
|---|---|---|---|---|
| Context Free | BCO | Convex | $\tilde{\mathcal{O}}\left(T^{3/4}/\varepsilon\right)$ | $\tilde{\mathcal{O}}\left(T^{3/4}/\varepsilon\right)$ [36] |
| | | Convex & Smooth | $\tilde{\mathcal{O}}\left(T^{2/3}/\varepsilon\right)$ | $\tilde{\mathcal{O}}\left(T^{3/4}/\varepsilon\right)$ [36] |
| | | S.C | $\tilde{\mathcal{O}}\left(T^{2/3}/\varepsilon\right)$ | $\tilde{\mathcal{O}}\left(T^{2/3}/\varepsilon\right)$ [36] |
| | | S.C & Smooth | $\tilde{\mathcal{O}}\left(T^{1/2}/\varepsilon\right)$ | $\tilde{\mathcal{O}}\left(T^{2/3}/\varepsilon\right)$ [36] |
| | MP-BCO | Convex | $\tilde{\mathcal{O}}\left(T^{1/2}/\varepsilon^2\right)$ | None |
| | | S.C | $\tilde{\mathcal{O}}\left(\log T/\varepsilon^2\right)$ | None |
| Context Based | Contextual Linear Bandits | | $\tilde{\mathcal{O}}(T^{3/4}/\varepsilon)$ | $\Omega(T)$ [34] |
| | Generalize Linear Bandits | | $\tilde{\mathcal{O}}(T^{3/4}/\varepsilon)$ | $\Omega(T)$ [34] |

**Reviewer 2:**
— "For contextual bandit, this work shows a sub-linear regret for LDP, but there exists a linear regret for DP..."
As explained in lines 53-61 in our submission ("collected information" there means the quantity based on use's private
data, like line 9 in Algorithm 3 without adding noise), our result doesn't contradict with the lower bound proved in
DP setting [34]. In more detail, on one hand, post-processing property holds only for the output of a DP algorithm
which doesn't use private data any more. However, in our algorithms for LDP contextual bandits, though we can use
post-processing property to prove estimation sequence $\{\tilde{\theta}_t\}$ satisfies DP, it doesn't imply the output action sequence
$\{x_t\}$ satisfies DP, as these actions are made in the local side which use private local data. On the other hand, to show the
difference more intuitively, assume the true parameter $\theta^*$ is known in advance. For two users with completely different
features, optimal actions for them should be different. In DP setting, it requires output actions to be close to each other,
which then will cause some regret inevitably. While in LDP setting, since we know $\theta^*$ and decisions are made locally
based on local features, we can choose optimal actions for each user locally without causing any regret. Hope these two
explanations help you understand our results and the difference with previous DP setting better.

— "the regrets for MP are not consistent, one is over epsilon squared, the other is over epsilon. ":
Thanks for pointing out this issue! It is a typo, and the regret for MP-BCO should be over $\epsilon^2$, since the regret in
non-private setting depends on $G^2$ ($G$ is the Lipschitz constant) which leads to $\epsilon^2$ here.

— "in the proof of Theorem 6, Eq(27) ... is not trivial":
Eq(27) holds after using the non-private guarantee $\text{Reg}(\mathcal{A}, \cdot)$ with respect to functions $\{\tilde{f}_t(x)\}$ defined in line 534.
Since the Lipschitz constant of $\{\tilde{f}_t(x)\}$ is upper bounded by $G + \sigma\sqrt{d}$, we obtain Eq(27) accordingly.

**Reviewer 3:**
— "Is the dependence on eps in the reduction optimal?":
We tend to believe our dependence on $\epsilon$ in the reduction is optimal in most cases, and there are some implications.
First, [Agarwal Singh 2017] achieved the form (non-private regret + $1/\epsilon$) under DP guarantee only for online linear
optimization rather than bandit setting, and [Tossou and Dimitrakakis 2015] achieved similar form under a variant of
DP focusing on a single output instead of regular DP focusing on sequential output. Besides, for MAB, [7] proved
lower bounds for several different versions of DP (including DP and LDP), which nearly match our upper bounds.

— "For the contextual setting, it is not accurate to say ... make the comparisons clear":
We agree with your comment! Actually in our submission, we do mention LDP and DP are not comparable in
contextual bandits (in lines 53-54) and only claim LDP is stronger than DP in context-free setting. Besides, LDP does
require some computational resource at users which may be unavoidable if we want to protect LDP and make accurate
recommendations simultaneously. We will explain all of these more clear in future version.

**Reviewer 4:**
— "The paper has no experiments. It is more interesting to see the trade-off ...":
Thanks for your suggestion! We have conducted some experiment for private MAB, and it indeed shows the trade-off
between privacy and utility. We will add experimental part in future version.

[Meta-Review · NeurIPS 2020]

After discussion, all reviewers agreed that there are interesting and novel contributions in this paper, and that it meets the bar for acceptance. The main outstanding concern is clarity, in particular with regard to how the novel LDP setting relates to the central DP setting for contextual bandits. (This is discussed in Review 3 and acknowledged in the author feedback.) More generally, I hope the authors can make the writing a bit easier to follow for the final version, following the suggestions of the reviewers.